# Neural Network-Based Identification of Cloud Types from Ground-Based Images of Cloud Layers

**Zijun Li** [1], **Hoiio Kong** [1,*] and **Chan-Seng Wong** [2]

1   Faculty of Data Science, City University of Macau, Macau 999078, China; lizj892112528@gmail.com
2   Macao Meteorological Society, Macau 999078, China; wbensonw@gmail.com
*   Correspondence: hikong@cityu.mo

**Abstract:** Clouds are a significant factor in regional climates and play a crucial role in regulating the Earth's water cycle through the interaction of sunlight and wind. Meteorological agencies around the world must regularly observe and record cloud data. Unfortunately, the current methods for collecting cloud data mainly rely on manual observation. This paper presents a novel approach to identifying ground-based cloud images to aid in the collection of cloud data. However, there is currently no publicly available dataset that is suitable for this research. To solve this, we built a dataset of surface-shot images of clouds called the SSC, which was overseen by the Macao Meteorological Society. Compared to previous datasets, the SSC dataset offers a more balanced distribution of data samples across various cloud genera and provides a more precise classification of cloud genera. This paper presents a method for identifying cloud genera based on cloud texture, using convolutional neural networks. To extract cloud texture effectively, we apply Gamma Correction to the images. The experiments were conducted on the SSC dataset. The results show that the proposed model performs well in identifying 10 cloud genera, achieving an accuracy rate of 80% for the top three possibilities.

**Keywords:** ground-based cloud images; cloud genera; identification; convolutional neural networks; texture; Gamma Correction

## 1. Introduction

Clouds are a significant element that affects regional climate, with different types of clouds reflecting varying amounts of sunlight based on their composition, altitude, and other factors, which can alter the Earth's climate [1,2]. As shown in Figure 1, under the influence of sunlight and wind, clouds play a significant role in the Earth's water cycle [3]. Local weather is closely linked to the type of cloud coverage [4], and, to some extent, future weather forecasts can be predicted based on the type of clouds.

Cloud types and structures are vital sources of meteorological information, and it is crucial for regional meteorological bureaus to consider the types and structures of clouds present in the area when compiling meteorological data and generating regional weather maps [5]. Accurate cloud data on barometric charts can aid the weather bureau in providing more precise weather forecasts. However, unlike other meteorological data that can be automatically collected (e.g., temperature, humidity, rainfall, etc.), cloud data still rely on manual observation and collection by observers. The Met Office requires hourly observations, or more frequent observations every half-hour for facilities that are significantly affected by weather and cloud conditions (e.g., airports), which leads to a significant increase in the cost of manually collecting cloud data. If we could achieve the automatic classification of cloud genera, it would greatly save the cost of collecting meteorological data. Currently, cloud classification is primarily performed manually by experts in the field. This method has the drawback of being time-consuming, labor-intensive, and relying solely on the expertise of academics for its accuracy, resulting in uncertainty in the classification results. An automated and precise classification method

would significantly enhance the consistency of cloud genera data collection for researchers. However, there is currently a lack of research on the automatic identification of cloud types from ground-based images.

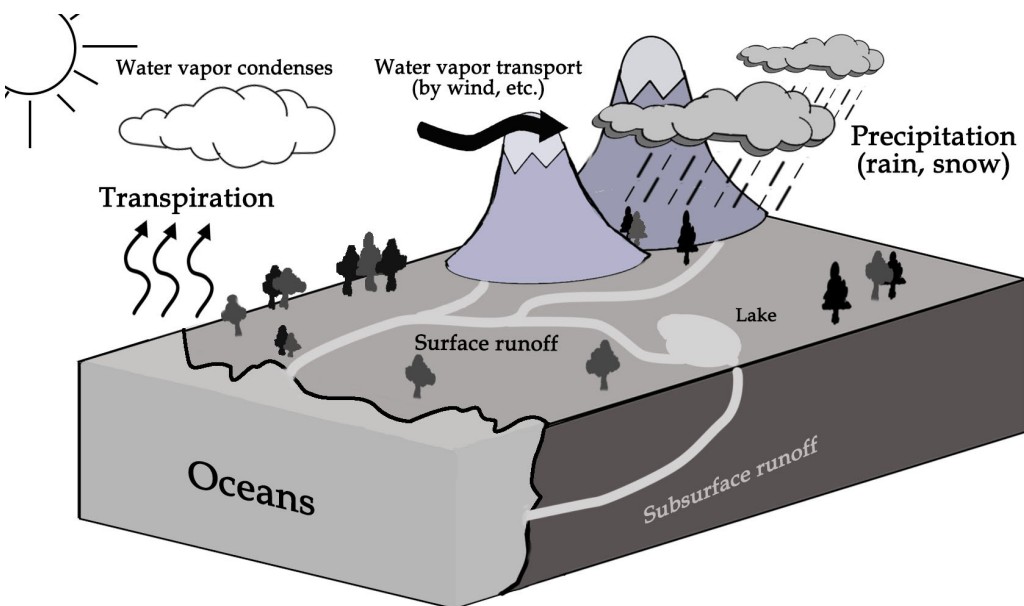

**Figure 1.** Diagram of the water cycle.

The current research on cloud classification has primarily been carried out by categorizing satellite images [6], which provide a direct view of cloud cover on the Earth's surface. However, this method is not highly effective in real-time as these satellites do not continuously monitor the same area, and the time required for data transmission is significant. In contrast, some research on cloud classification has mainly utilized cloud images captured from the ground to identify cloud genera. Researchers have proposed using the textural structure of infrared images [7] to perform cloud classification, focusing on features such as mean grey level, edge sharpness, and cloud gaps. While these methods perform well overall, they may not be accurate enough for identifying similar clouds. This is why some research considers adding the spectral information of clouds as features [8]. While fewer studies have utilized ground-based images for classification, this approach is still feasible. In fact, some studies have employed support vector machines (SVM) [9] to classify clouds based on their texture and structural features. Nevertheless, this approach demands high-quality images and may impact classifier performance if the images are captured during nighttime or under conditions of rain or snow. Along with improvements in camera technology, the use of whole-sky images [10] and infrared cloud analysis instruments [11] provide the basis for collecting high-quality ground cloud image data.

The progress of machine learning has led to the widespread use of artificial neural networks (ANN) in the field of computer vision (CV), resulting in significant achievements [12]. Numerous scholars are attempting to incorporate ANNs in meteorology, with convolutional neural networks (CNN) garnering significant attention and producing remarkable results [13–15]. Presently, some researchers have also started to employ CNN as a technique for cloud image identification. Traditional CNN operates as a hierarchical framework for feature extraction. In a CNN, the lower layers extract the cloud's texture (e.g., structure, fringes, etc.), whereas the higher layers capture more complex semantic information. Despite ground-based images having limitations in representing spectral information and cloud composition, the texture features of clouds remain the primary basis for their classification. Ground-based images are better suited to depict this feature [9]. Considering the potentially subtle textural variations between cloud genera in the implementation, CNNs' ability to efficiently extract feature information through convolutional

kernels offers a means to achieve the automatic and precise classification of ground-based cloud images. The DeepCloud network was the first to attempt to use neural networks to identify ground-based cloud images [16]. The researchers classified clouds into eight categories and supplemented them with clear sky images. They then employed deep neural networks to extract features from the images and achieved an accuracy of 80%, demonstrating the efficacy of neural network classification. Following the DeepCloud study, subsequent research has also utilized neural networks to identify ground-based cloud images, known as the CloudNet [17]. In the CCSN database, clouds were classified into 11 categories, and the model was enhanced by normalizing the RGB three-channel values in the images and increasing the image contrast, ultimately resulting in a high level of accuracy. Subsequently, Huertas-Tato et al. proposed combining neural networks and random forest to further improve the classification performance of the model [18]. Shuang et al. proposed MMFN, which could learn extended cloud information by fusing heterogeneous features in a unified framework [19]. TGCN is a model that learns features in a supervised manner and incorporates graph computation into the model [20]. However, these studies have not addressed the problem of accurately classifying similar cloud formations. In summary, there are few studies in the fields of meteorology or image recognition that use neural networks to identify ground-based cloud images, and all these studies encounter the challenge of having inadequate data samples or poor data quality. In response to the above problems, we built a ground-based cloud image dataset and proposed a method based on CNN to identify ground-based cloud images. The main contributions of this research are as follows:

1.  Constructing a ground-based cloud images dataset, known as the SSC, which is the first dataset to ensure roughly equal sample sizes for each cloud genera. The dataset comprises 1092 data samples, categorized into 10 groups.
2.  Various processing methods are employed to adjust the contrast and brightness of the images, thereby enhancing the texture of the clouds. By this approach, the efficiency of the model in extracting cloud texture is enhanced, and the model's performance is improved.
3.  We employed a new evaluation method for assessing the performance of our model. The method involves categorizing clouds into three types based on their texture (cumulus, stratiformis, and undulatus). While this evaluation method is relatively uncommon in machine learning, it is crucial in the field of meteorology. In reality, these three types of clouds have varying degrees of impact on the climate, which is an important consideration when collecting cloud data [21,22].

The remainder of this paper is structured as follows: Section 2 provides a brief introduction to our dataset; Section 3 details a method for improving the recognition of ground-based cloud images by the model through image enhancement; Section 4 presents the experimental results and provides a discussion on the feasibility and effectiveness of the proposed approach; Section 5 is where a summary of the research is presented, along with a discussion on future research directions to explore the practical applications of the results.

## 2. Data

At present, the International Cloud Atlas (WMO) [23] classifies clouds into three families and ten genera according to their height and textures. In addition, according to the different reasons for cloud formation, they can be divided into cumulus, stratiformis, and undulatus, as shown in Table 1 and Figure 2, which enumerate each cloud sample in the datasets. Clouds generated by convective motion are cumulus, clouds formed by systematic vertical motion are stratiformis, and clouds formed by atmospheric fluctuations or atmospheric turbulence are undulatus. However, no meteorological office or research institute has a dedicated database of ground-based cloud maps yet. Therefore, this study will use CCSN [17] and the dataset from the Cloud Watching Contest [24] as a foundation to establish a better dataset and improve it under the supervision of the Macau Meteorological Society. We then removed poor-quality images from the dataset (e.g., images containing

a certain proportion of buildings or trees). Specifically, we eliminated data samples with less than a certain proportion of clouds in a 480 × 480-pixel image. Both the CCSN and the Cloud Watching Contest datasets were obtained from a single institution, indicating that all images in the same dataset were captured at the same location. Additionally, cloud production is influenced by geographic location, cold, and warm currents [25]. Therefore, if data were captured from the same location, the number of the same cloud genera in the dataset would be excessively high. To avoid having one category of data samples significantly outnumber the others and causing the model to overly focus on that category during feature extraction, thereby affecting the overall performance of the model, we additionally collected and labeled cloud genera images with a smaller sample size and added them to the SSC. This ensures that the data volume of each cloud genera in the dataset is roughly the same. The types of cloud genera in our dataset and the amount of data samples are shown in Table 1.

**Table 1.** Classification and quantity of cloud genera.

| Heigh | Cloud Genera | Texture | Quantity |
|---|---|---|---|
| High-level | Cirrus (Ci) | Stratiformis | 103 |
| | Cirrocumulus (Cc) | Undulatus | 103 |
| | Cirrostratus (Cs) | Stratiformis | 110 |
| Middle-level | Altostratus (As) | Stratiformis | 100 |
| | Altocumulus (Ac) | Undulatus | 106 |
| Low-level | Nimbostratus (Ns) | Stratiform | 133 |
| | Stratus (St) | Undulatus | 102 |
| | Stratocumulus (Sc) | Undulatus | 101 |
| | Cumulus (Cu) | Cumulus | 107 |
| | Cumulonimbus (Cb) | Cumulus | 127 |
| Total | | 1092 | |

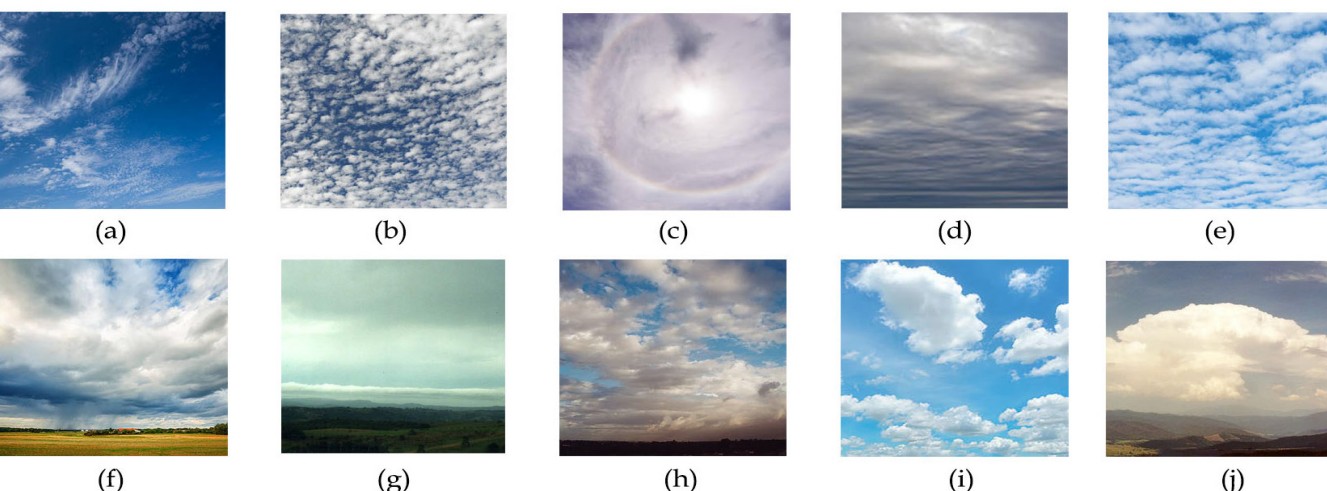

**Figure 2.** Display of the samples of each category in the SSC dataset. (**a**) Ci; (**b**) Cc; (**c**) Cs; (**d**) As; (**e**) Ac; (**f**) Ns; (**g**) St; (**h**) Sc; (**i**) Cu; (**j**) Cb.

## 3. Method

### 3.1. Image Processing

The classification of ground-based cloud images is primarily based on the textural characteristics of the clouds. However, some cloud genera have less distinctive textural characteristics than others, and the differences between them are mainly based on their height and composition. For example, Altocumulus (Ac) and Cirrocumulus (Cc) share a similar appearance of a patch, sheet, or layer of cloud. However, the distinguishing factor

is that Cc has a white cloud base with a ripple-like texture, while Ac has a greyish cloud base with smaller and well-defined cloud patches. It may be challenging to accurately differentiate between cloud genera with similar textural characteristics based solely on the original images. To emphasize these textural characteristics for subsequent feature extraction, we will pre-process all data samples. There are two methods: edge detection and contrast enhancement.

Edge detection is typically carried out using the Sobel operator [26], which suppresses discrete differences by computing a grey-scale approximation of the image. This algorithm employs two 3 × 3 convolution kernels to convolve in the x and y directions, respectively, and the convolution kernels are

$$G_x = \begin{bmatrix} -1 & 0 & 1 \\ -1 & 0 & 1 \\ -1 & 0 & 1 \end{bmatrix}, G_y = \begin{bmatrix} -1 & -1 & -1 \\ 0 & 0 & 0 \\ 1 & 1 & 1 \end{bmatrix} \quad (1)$$

The image is convolved and the approximate gradient value of each pixel in the image is calculated $g$:

$$|G| = |G_x| + |G_y|, \quad (2)$$

$$g = \begin{cases} |G|, |G| \geq \rho \\ 0, |G| < \rho \end{cases} \quad (3)$$

Typically, algorithms aiming for efficiency in the future will not directly use $G_x$ and $G_y$, but will calculate them using the approximate values of their square roots. Then, where $\rho$ is the hyperparameter, and when the grey gradient value $g$ is bigger than $\rho$, it will be considered as graphical edge retention; otherwise, it will be suppressed. The Sobel operator detects edges by assigning weights to the differences based on the grey values of neighboring pixels surrounding pixel point 8. This phenomenon reaches an extreme value at the edge. The operator also has a smoothing effect on noise and provides relatively quick and accurate information on the direction of edges. However, it is not very precise in locating the edges.

Edge detection will also be performed using the Canny operator [27], which firstly smooths the image through a Gaussian filter with a Gaussian kernel *G*:

$$G(x, y) = \frac{1}{\sqrt{2\pi}\sigma} e^{-\frac{x^2 + y^2}{2\sigma^2}}, \quad (4)$$

where $\sigma$ is the hyperparameter that determines the width of the Gaussian function. The magnitude and direction of the gradient are then calculated. The Canny operand uses two convolution kernels in the x and y directions to calculate the edge pixels in the *x*, and *y* directions, using the convolution kernel in the Sobel operator, as shown in Equation (1). In this case, $G_x$ performs edge detection in the x-direction, and when the convolution kernel calculates the boundary pixels perpendicular to the x-direction, it will zoom in on the value of the pixel point on the boundary; $G_y$ performs edge detection in the y-direction, and the operation is the same as $G_x$. The gradient amplitude is later calculated by the above convolution kernel:

$$Edge_{Gradient(G)} = \sqrt{G_x^2 + G_y^2}, \quad (5)$$

and gradient direction calculation:

$$Angle(\theta) = tan^{-1}\left(\frac{G_y}{G_x}\right) \quad (6)$$

During the Gaussian filtering process, it is possible that the edge pixels of the image may be enlarged. Therefore, after completing the Gaussian filtering, it is necessary to filter out the non-edge pixel points using non-maximum suppression. This helps to ensure that

the width of the edge is kept to a minimum of one pixel. If a pixel is an edge of the graph, it will have the maximum gradient value in the gradient direction; otherwise, the grey value is set to zero, which helps to refine the graph edge. The Canny algorithm performs non-maximum suppression in four directions:

1. $\theta = 0^{\circ}$, contrast the left and right neighboring points of the pixel.
2. $\theta = 45^{\circ}$, contrast the lower-left with the upper-right neighborhoods of the pixel.
3. $\theta = 90^{\circ}$, contrast the top and bottom neighborhoods of the pixel.
4. $\theta = 135^{\circ}$, contrast the upper-left with the lower-right neighborhoods of the pixel.

Finally, using the hysteresis thresholding method, the hysteresis threshold requires two thresholds (high and low threshold). As shown in Figure 3, an edge pixel with a gradient value higher than the high threshold is labeled as a strong edge pixel; an edge pixel with a gradient value less than the high threshold and greater than the low threshold is labeled as a weak edge pixel; and an edge pixel is suppressed if its gradient value is less than the low threshold. The Canny operator is not susceptible to noise interference and is able to detect the true weak edges. Moreover, the Canny operator excels at locating edge pixels and accurately identifying edge points on pixels that have the highest grayscale variation, but the Canny operator's computational complexity is significantly higher, which reduces its efficiency.

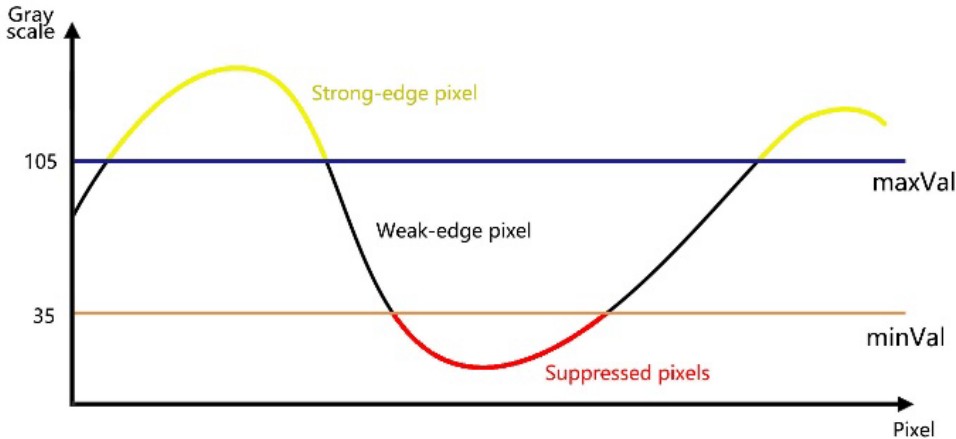

**Figure 3.** Displaying the hysteresis threshold, when pixel values greater than maxVal are flagged as strong edges; pixel values less than maxVal and greater than minVal are flagged as weak edges; pixel values less than minVal are suppressed.

Contrast enhancement uses the Clahe algorithm [28], which is further improved by the Histogram Enhancement algorithm (HE). This method enhances the contrast of an image by locally performing histogram equalization, which preserves more details of the image and facilitates the extraction of features. The Clahe algorithm first adjusts the luminance distribution of pixels by equalizing the histogram, while it avoids the issue of certain colors overpowering others by limiting the number of pixels per color. By comparing the histogram of the original image in part a of Figure 4 with the histogram processed by the Clahe algorithm in part b, it is evident that the Clahe operator alters the grey values of pixels that exceed a certain threshold to a lower value. By reducing the brightness of the bright areas and increasing the brightness of the dark areas, the Clahe algorithm enhances the contrast of the image.

The image enhancement method still utilizes Gamma Correction [29], a non-linear operation that adjusts the brightness of the image. The corresponding pixel value is determined using a transformation formula, which is represented as

$$s = cr^{\gamma} \tag{7}$$

where *r* is the input pixel grey value, *c* is a constant, usually taken as 1, and $\gamma$ is the Gamma Correction value, which determines whether the output pixel grey value s is transformed to a larger or smaller value. As shown in Figure 5, when $\gamma > 1$ this transformation is called decoding gamma, and the final image will be darker overall; when $\gamma < 1$, this transformation is called coding gamma, and the final image will be brighter overall.

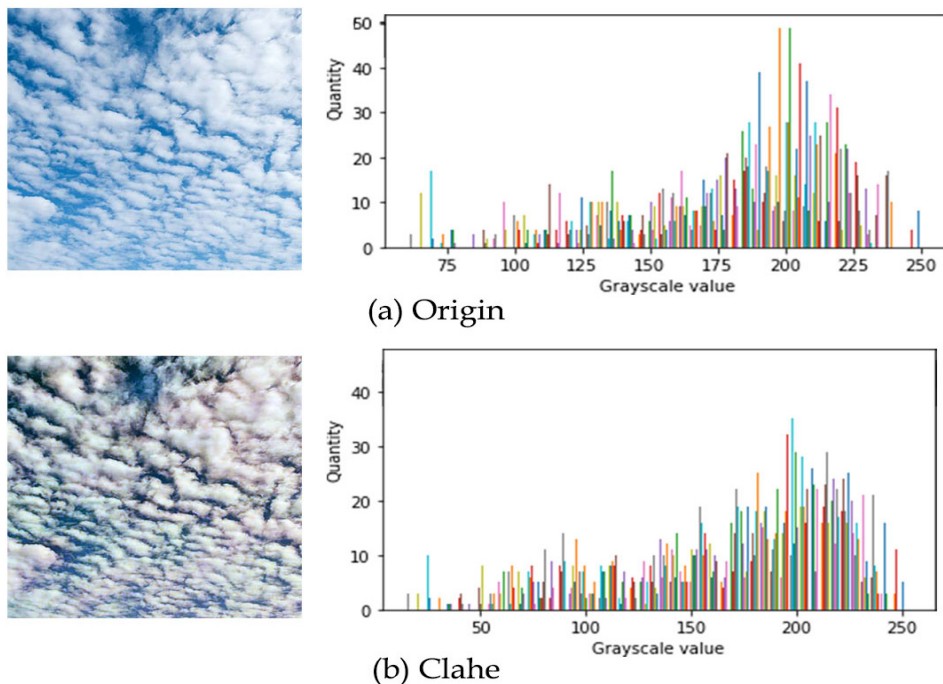

(a) Origin

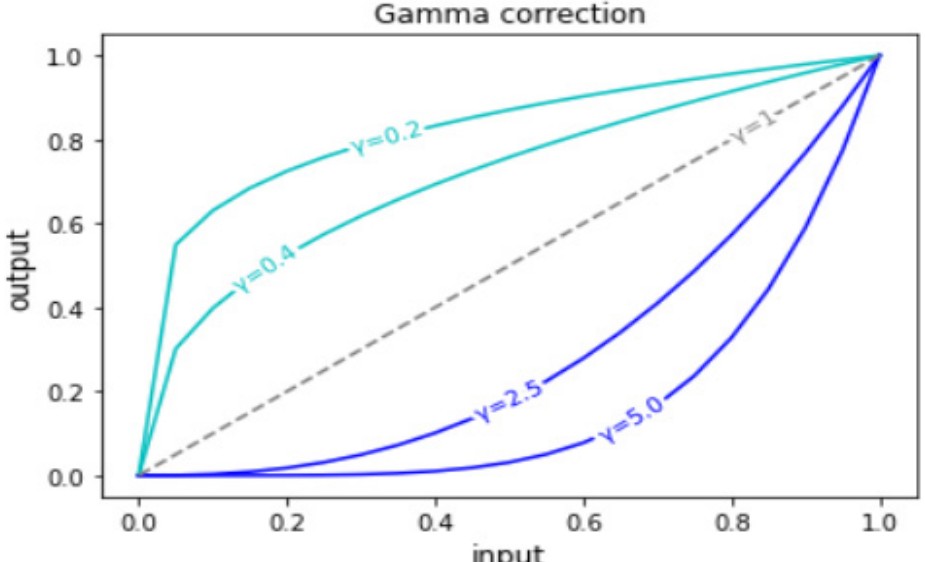

(b) Clahe

**Figure 4.** (**a**) The original image and its grayscale histogram. (**b**) The image is processed by Clahe and its grayscale histogram.

**Figure 5.** Gamma Correction for different values of $\gamma$.

We utilize the above four methods to enhance the original data, aiming to strengthen the texture information present in the image. The results obtained after pre-processing the images are presented in Figure 6. We will utilize the four datasets obtained after processing the original dataset to train the models and compare their performance.

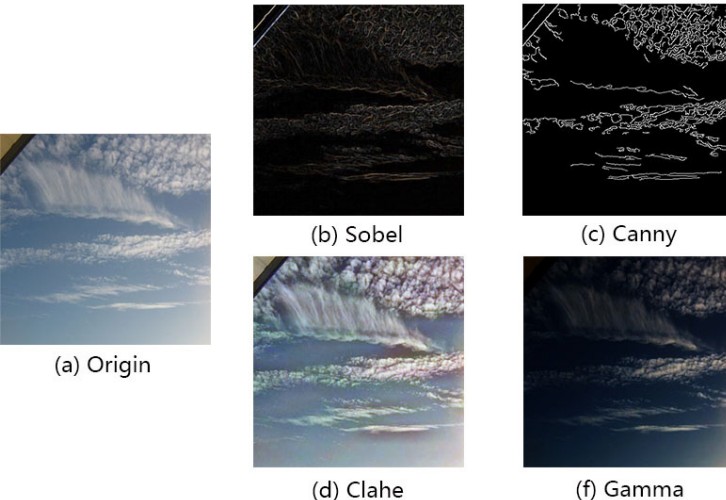

**Figure 6.** Original image and processed images.

*3.2. Model*

Neural networks (NNs) are a classic field in the domain of machine learning. In NNs, problems such as classification discrimination are solved by computing non-linear functions in neurons, as illustrated in Figure 7. A classical NN network contains an input layer, one or more hidden layers, and an output layer. This structure represents the underlying multilayer perceptron (MLP) structure [30,31]. However, NNs face a challenge due to their fully connected network structure, which requires them to handle a large number of operations between their hidden layers. In the field of image recognition, NNs treat the data as a matrix vector and extract feature values from it. The feature vector is then passed to all the neurons in the next layer, which repeat the previous step until the data are passed to the output layer. This means that if we attempt to increase the number of image channels, the total number of pixels, or add new neurons to the hidden layer, the amount of computation required will increase significantly. As a result, the resulting increase in time and memory consumption is unacceptable. Compared to traditional NN, the CNN proposed by LeCun et al. addresses this problem to some extent [32–34]. In a CNN, not all neurons in the upper and lower layers are connected directly. Instead, they are partially connected through a medium known as 'convolutional kernels', which is added to the hidden layer. Simultaneously, a convolutional kernel is shared between neurons in the same layer, which retains the location information of features in the image. As the data are passed down the convolutional layer, the model extracts increasingly complex texture features. CNN is particularly effective in extracting local features from an image through convolution kernels, which can limit the number of parameters in the model. This makes CNN more suitable for the field of image recognition. In addition, CNN also includes pooling layers (also known as subsampling or downsampling), which reduce the dimensionality of the image by performing a streamlining operation (e.g., maximization, averaging, etc.) on the data within a certain region, retaining only the key information.

There are numerous classical network structures in CNN (e.g., Inception [35] proposed by Google, etc.). The CNN employed in this paper is the classical VGG16 network structure [36], which belongs to a category of models that are built on the enhancements proposed by AlexNet [37]. Compared to AlexNet, the VGG network structure contains a greater number of convolution and pooling layers, utilizing smaller convolutional kernels such as $3 \times 3$ or $1 \times 1$. This enables the model to learn subtle differences in texture features during training, making it well-suited for extracting and distinguishing features among various cloud types. As shown in Figure 8, VGG16 consists of 16 convolutional layers and 5 pooling layers. Finally, the data are fed into a Fully Connected layer. Dropout layers [38] were added between the Fully Connected layers and Softmax layers as one of the methods to prevent overfitting of the model. The output of the final softmax layer corresponds

to the probability distribution of the 10 classification categories. During training, Adam optimizers [39] were used, which limit the update step size to an approximate range, automatically adjust the learning rate, and have a good interpretation of the hyperparameters. In summary, the network was fed images of size 480 × 480 as RGB trichrome channel values, and the output of the network corresponded to the probability distribution over 10 cloud genera labels.

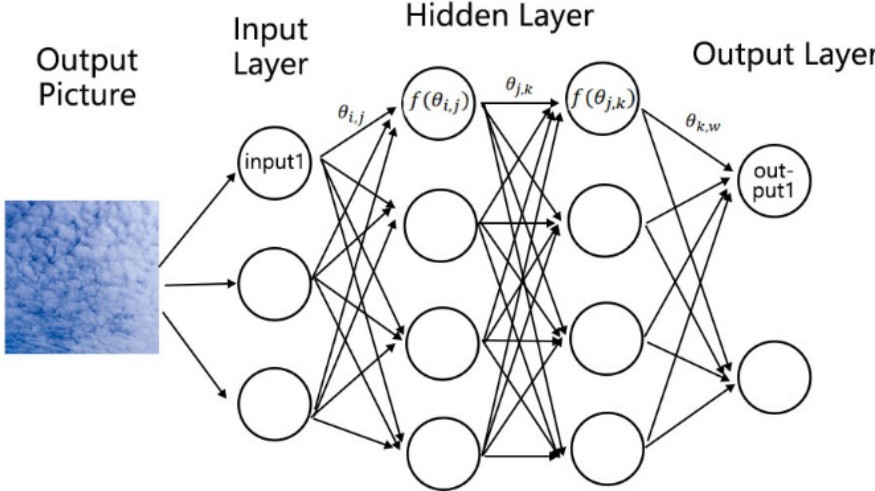

**Figure 7.** In the traditional neural network (NN) architecture, a node represents a neuron, $\theta$ represents the weight parameter of the layer, and $f(\theta)$ represents the activation function used by the neuron. NN is fully connected, where each neuron is connected to all neurons in the next layer.

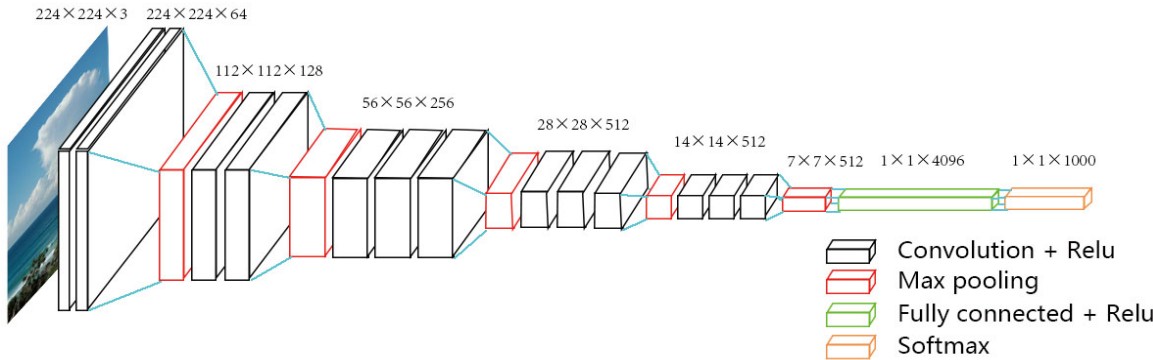

**Figure 8.** VGG16 structure. The network contains 16 convolutional layers, 5 pooling layers, and 1 fully connected layer.

## 4. Results and Discussion

To verify the validity of the proposed method in this paper, we evaluated the performance of the model on the 10 cloud genera using various methods:

1. Whether the first three items of the classification result include the correct labels.
2. Classification by cloud texture (divided into Stratiformis, Undulatus, Cumulus).

Refer to Table 1 for the specific method of classifying clouds based on their texture. In addition, we assessed the precision rate, recall rate, and F-measures for each evaluation method. Accuracy is concerned with the correctness of the objects classified in the model and is not dependent on whether the objects are fully labeled or have a high classification threshold. Recall is concerned with whether the model labels all the objects that belong to a certain class, regardless of whether other objects are misclassified. It is typically associated with a low classification threshold. The evaluation metrics are calculated based

on True Positive (*tp*), True Negative (*tn*), False Positive (*fp*), and False Negative (*fn*) for the classification results generated by the model, as shown in Equations (8) and (9).

$$Precision = \frac{tp}{tp + fp}, Recall = \frac{tp}{tp + fn} \tag{8}$$

$$
\begin{aligned}
F - score(\beta) &= \frac{(1+\beta^2)Precision*Recall}{\beta^2*Precison+Recall} \\
&= \frac{(1+\beta^2)tp}{(1+\beta^2)tp+\beta^2 fp+fn}
\end{aligned}
\tag{9}
$$

where $\beta$ is the hyperparameter, usually set to 1. The result is considered optimal when the $F - score(\beta)$ value is 1; 0 is considered the worst result.

In this paper, we used the Sobel, Canny, Clahe, and Gamma methods, respectively, to process the SSC and generate four new datasets. These datasets were applied to the VGG16 model (Figure 8), and the classification results were evaluated using the precision rate, recall rate, F-measures, and accuracy mentioned above. The results indicated that the Gamma Correction dataset performed best, as shown in Table 2. The majority of cloud genera displayed high classification recall of 80% and precision. However, the classification results of Cirrus (Cs) were relatively poor. Upon examining the model's classification 20 times, the Cs labeled data sample model classification was shown in Figure 9, where the majority of misclassified results were identified as Altocumulus (As). It is because both cloud genera belong to Stratiformis and share high similarity in texture features, and the International Cloud Atlas distinguishes between them primarily based on the height of the cloud base and differences in composition. Based on the classification of cloud texture features, all three categories displayed over 70% correctness in both recall and precision. This indicates that the model has a strong discriminatory ability for cloud texture features.

**Table 2.** Performance evaluation of Gamma datasets with multiple classification methods.

| Categorty | Recall | Precision | F1 | Accuracy |
|---|---|---|---|---|
| AC | 0.75 | 0.83 | 0.79 | |
| AS | 0.8 | 0.64 | 0.71 | |
| CB | 0.85 | 0.81 | 0.83 | |
| CC | 1 | 0.77 | 0.87 | |
| CI | 0.6 | 1 | 0.75 | 0.78 |
| CS | 0.5 | 0.67 | 0.57 | |
| CU | 0.9 | 0.67 | 0.77 | |
| NS | 0.8 | 0.76 | 0.78 | |
| SC | 0.75 | 0.94 | 0.83 | |
| ST | 0.85 | 0.89 | 0.87 | |
| Stratiformis | 0.675 | 0.74 | 0.71 | |
| Undulatus | 0.675 | 0.68 | 0.68 | 0.71 |
| Cumulus | 0.85 | 0.71 | 0.77 | |

As shown in Table 3, the Clahe dataset has the second-best performance following the Gamma dataset. While the overall performance of Clahe in identifying cloud species is slightly better than Gamma, precision analysis shows that some cloud genera had high precision while others had low precision. There was a significant difference in the performance between the two parts of cloud genera. Our belief is that the Clahe dataset does not showcase cloud textures as effectively as Gamma, which is supported by its inferior performance in texture feature classification compared to Gamma. Upon comparing the recall of both, we observed that Stratiformis and Cumulus have significantly higher recall rates in Gamma compared to Clahe. Additionally, we found that several cloud genera had a high recall but low precision in Clahe. This finding further supports the conclusion that Clahe is less effective in processing cloud texture features compared to Gamma. From Tables 4 and 5, it can be observed that the overall performance of the Sobel dataset is

similar to that of the Canny dataset, with both performing worse than Clahe. We found that after processing with Canny, the model struggled to identify Stratus (St) and was likely to incorrectly classify cloud genera as Nimbostratus (Ns). Furthermore, Canny performed worse in classifying cloud genera based on texture, with an accuracy of only 54%. It struggled to differentiate Cumulus from the other two categories.

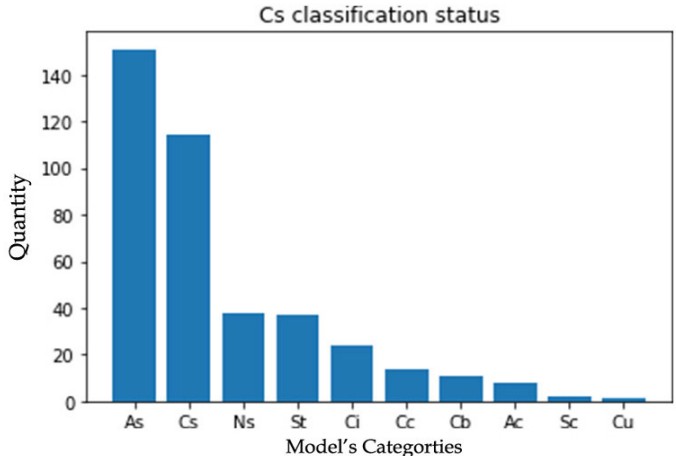

**Figure 9.** The situation of Cirrus (Cs) labeled samples under 20 times performed by the model when using Gamma dataset.

**Table 3.** Performance evaluation of Clahe datasets with multiple classification methods.

| Categorty | Recall | Precision | F1 | Accuracy |
|---|---|---|---|---|
| AC | 0.9 | 0.6 | 0.72 | |
| AS | 0.7 | 0.74 | 0.72 | |
| CB | 0.8 | 0.9 | 0.85 | |
| CC | 0.95 | 0.9 | 0.93 | |
| CI | 0.55 | 0.85 | 0.67 | |
| CS | 0.95 | 1 | 0.97 | 0.8 |
| CU | 0.9 | 0.67 | 0.77 | |
| NS | 0.65 | 0.8 | 0.72 | |
| SC | 0.85 | 0.65 | 0.74 | |
| ST | 0.95 | 0.70 | 0.83 | |
| Stratiformis | 0.54 | 0.77 | 0.63 | |
| Undulatus | 0.77 | 0.63 | 0.69 | 0.65 |
| Cumulus | 0.78 | 0.76 | 0.77 | |

**Table 4.** Performance evaluation of Sobel datasets with multiple classification methods.

| Categorty | Recall | Precision | F1 | Accuracy |
|---|---|---|---|---|
| AC | 0.65 | 1 | 0.79 | |
| AS | 0.8 | 0.5 | 0.62 | |
| CB | 0.85 | 0.77 | 0.81 | |
| CC | 0.7 | 1 | 0.82 | |
| CI | 0.75 | 0.56 | 0.64 | |
| CS | 0.6 | 0.92 | 0.73 | 0.72 |
| CU | 0.8 | 0.1 | 0.86 | |
| NS | 0.95 | 0.95 | 0.89 | |
| SC | 0.75 | 1 | 0.88 | |
| ST | 0.85 | 0.57 | 0.68 | |
| Stratiformis | 0.72 | 0.63 | 0.67 | |
| Undulatus | 0.54 | 0.61 | 0.57 | 0.64 |
| Cumulus | 0.65 | 0.68 | 0.67 | |

**Table 5.** Performance evaluation of Canny datasets with multiple classification methods.

| Categorty | Recall | Precision | F1 | Accuracy |
|---|---|---|---|---|
| AC | 0.9 | 0.9 | 0.9 | |
| AS | 0.95 | 0.34 | 0.5 | |
| CB | 0.8 | 1 | 0.89 | |
| CC | 0.6 | 1 | 0.75 | |
| CI | 0.85 | 1 | 0.92 | 0.71 |
| CS | 0.7 | 1 | 0.82 | |
| CU | 0.65 | 0.76 | 0.7 | |
| NS | 0.55 | 0.67 | 0.6 | |
| SC | 1 | 0.29 | 0.45 | |
| ST | 0 | 0 | 0 | |
| Stratiformis | 0.56 | 0.56 | 0.56 | |
| Undulatus | 0.66 | 0.54 | 0.59 | 0.54 |
| Cumulus | 0.25 | 0.5 | 0.33 | |

## 5. Conclusions

In this paper, we propose a CNN-based method for identifying 10 cloud genera. Due to the reliance on the identification of the texture of the clouds, the images undergo pre-processing to enhance these features. The classical VGG16 architecture of the CNN model is then employed for training. We have also constructed a dataset of high-quality cloud images, known as the SSC. Moreover, the SSC is the first dataset to ensure an approximately equal number of data samples for all 10 cloud genera.

Following multiple rounds of experimentation, the results demonstrate that the model presented in this paper can accurately identify cloud genera using ground-based cloud images and performs well on our newly established database. The model has some misclassification errors, mainly focused on As and Cs. This is due to the similarity in their texture features, as well as a transformation phenomenon, making it challenging to discriminate based on image texture alone. Additionally, cloud height is currently a critical indicator for distinguishing between these two cloud genera. We have also demonstrated that the model's performance can be optimized by adjusting the brightness or darkness of the images.

At present, our research on cloud identification solely relies on cloud texture. However, there are numerous features that can be utilized to identify clouds, such as cloud height and composition. For the future, we are considering incorporating hardware equipment (e.g., the ceilometer) to enable us to collect cloud height data while capturing cloud images. We hope to address the difficulty in distinguishing between As and Cs by adding the feature of cloud height and improving the performance of the model. In addition, multiple cloud genera can often appear in the same sky simultaneously. In the next stage, we plan to implement distinguishing between different cloud genera in the same picture. Our research results have practical applications. For example, meteorological organizations can utilize our method to collect cloud data in unpopulated areas (e.g., plateaus, oceans, etc.). On the other hand, our method is highly real-time, allowing meteorological bureaus to monitor clouds in real-time using camera equipment to forecast future local weather changes. Furthermore, our method enables the public, without prior meteorological knowledge, to identify between different cloud genera.

**Author Contributions:** Conceptualization, Z.L. and H.K.; methodology, Z.L.; software, Z.L. and H.K.; investigation, H.K., Z.L. and C.-S.W.; resources, C.-S.W.; writing—original draft preparation, Z.L.; writing—review and editing, H.K. and Z.L.; visualization, Z.L.; supervision, H.K. and C.-S.W.; project administration, H.K.; funding acquisition, H.K. All authors have read and agreed to the published version of the manuscript.

**Funding:** This research was supported by Macau Foundation under its Research Fund (Grant No. MF2102), Macau.

**Institutional Review Board Statement:** Not applicable.

**Informed Consent Statement:** Not applicable.

**Data Availability Statement:** The data presented in this study are available on request from the corresponding author.

**Acknowledgments:** We thank the Macao Meteorological Society for their support in the data checking part.

**Conflicts of Interest:** The authors declare no conflict of interest.

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
