# Peer review of "Neural Network-Based Identification of Cloud Types from Ground-Based Images of Cloud Layers"

_applsci, doi:10.3390/app13074470_

Round 1

Reviewer 1 Report

The study has some really lofty goals, and based on what I've read, its authors want to make a significant advancement in a field that is both very important and connected to the dynamics of a natural system. This is a very encouraging sign. The authors, unfortunately, do not give a foundational understanding of "cloud production," which is related to the assertion stated in the first paragraph of the Introduction of the work, which states that "Clouds are a major element determining regional climate." Hence, there is a need for a diagram to show the formation of the cloud system as well as how the images generated within the system can be used for further analysis. This is one of the major concerns I have with the paper. In spite of the fact that I believe the paper is deserving of publication, I still have some other significant issues, which are as follows:

1. The intention of the final paragraph of the "Introduction" is to provide the contribution of the paper, which is a good practise; however, it is complicated in the sense that it does not either present the contributions of the work clearly, nor does it tend to be as if the authors are presenting the contents of the section paper. This is a problem because it is a good practice to provide the contribution of the paper in the second to the last paragraph of the Introduction section, while the last paragraph of the introduction section can be the outline of the paper.

2. There should be a separate section titled "Literature Review," which is supposed to be section 2. In this section, all of the related work needs to be brought up, and it should let the reader get a clear idea of what is happening in the area of the study and how you draw your research gaps. This section should be separate.

3. Although the experimental analysis is sound, the presentation of the findings is not at all satisfactory. The absence of Table 2 is one of the most significant problems.

4. In conclusion, there is a requirement for a discussion on the consequence of classification with regard to the Model that was applied in addition to the result that was attained.

Author Response

Point 1: The intention of the final paragraph of the "Introduction" is to provide the contribution of the paper, which is a good practise; however, it is complicated in the sense that it does not either present the contributions of the work clearly, nor does it tend to be as if the authors are presenting the contents of the section paper. This is a problem because it is a good practice to provide the contribution of the paper in the second to the last paragraph of the Introduction section, while the last paragraph of the introduction section can be the outline of the paper.

Response 1: Thank you very much for the comment. We have revised the final paragraph of the introduction and listed the contributions of this paper in bullet points. Additionally, a paragraph has been added to introduce the outline of the paper in the introduction.

Point 2: There should be a separate section titled "Literature Review," which is supposed to be section 2. In this section, all of the related work needs to be brought up, and it should let the reader get a clear idea of what is happening in the area of the study and how you draw your research gaps. This section should be separate.

Response 2: In the introductory section, we have reworked the section about related work. It highlights the use of satellite cloud maps as a mainstream method for identifying cloud genera and ongoing research in the field of machine learning for cloud genus identification based on ground images. We conclude by summarizing the current limitations of such research.

Point 3: Although the experimental analysis is sound, the presentation of the findings is not at all satisfactory. The absence of Table 2 is one of the most significant problems.

Response 3: We have added Table 2.

Point 4: In conclusion, there is a requirement for a discussion on the consequence of classification with regard to the Model that was applied in addition to the result that was attained.

Response 4: In conclusion, we propose directions for future improvements and outline specific ways in which our results can be applied.

Reviewer 2 Report

The content studied in the manuscript is very interesting and has very practical applications. The reviewers highly appreciated it. The authors propose a convolutional neural network-based approach to classify images of clouds taken from the ground. This method improves the accuracy of cloud genus classification and avoids the problem of relying solely on the experience of academics to choose the classification. Also, the authors build a more complete dataset of cloud images. Compared with the previous dataset, this dataset has more accurate classification and more uniform data samples for each cloud species.

Overall, the manuscript is worthy of publication, after the following issues have been addressed:

1.The state of research on recognition using neural networks is not sufficiently presented. The reviewer believes that someone must have conducted corresponding research in cloud classification recognition, please expand it in the introduction.

2.The tables in the full paper are not standardized and should be laid out as a three-line table.

3.Figure 7 is very blurred and the clarity does not meet the publication requirements, and needs to be revised.

4.In terms of the amount of content, there are not many contents and figures in the whole paper, so we suggest the authors to elaborate the specific conclusions obtained as well as the shortcomings of the study and the future related research outlook in the conclusion.

5.In terms of research object, the manuscript has a strong application value, but the scientific innovation is not very clear, so the authors are requested to add it in the introduction.

6.The following literature on the application of neural networks is suggested to be cited in the introduction

Qiang Li, Hongtao Jia, Qing Qiu, Yongzhu Lu, Jun Zhang, Jianghong Mao, Weijie Fan, M. F. Huang. Typhoon-induced fragility analysis of transmission towers in Ningbo area considering the effects of long-term corrosion [J]. Applied Sciences, 2022, 12(9), 4774.

Qiang Li, Hongtao Jia, Jun Zhang, Jianghong Mao, Weijie Fan, M. F. Huang, Bo Zheng. Typhoon loss assessment in rural housing in Ningbo based on township-level resolution [J]. Applied Sciences, 2022, 12(7), 3463.

Author Response

Response to Reviewer 2 Comments

Point 1: The state of research on recognition using neural networks is not sufficiently presented. The reviewer believes that someone must have conducted corresponding research in cloud classification recognition, please expand it in the introduction.

Response 1: Thank you. In the introductory section, we have reworked the section about related work. It highlights the use of satellite cloud maps as a mainstream method for identifying cloud genera and ongoing research in the field of machine learning for cloud genus identification based on ground images. We conclude by summarizing the current limitations of such research.

Point 2: The tables in the full paper are not standardized and should be laid out as a three-line table.

Response 2: We have modified the tables to make them standardized.

Point 3: Figure 7 is very blurred and the clarity does not meet the publication requirements, and needs to be revised.

Response 3: We redraw Figure 7 to make it clearer.

Point 4: In terms of the amount of content, there are not many contents and figures in the whole paper, so we suggest the authors to elaborate the specific conclusions obtained as well as the shortcomings of the study and the future related research outlook in the conclusion.

Response 4: In conclusion, we propose directions for future improvements and outline specific ways in which our results can be applied.

Point 5: In terms of the amount of content, there are not many contents and figures in the whole paper, so we suggest the authors to elaborate the specific conclusions obtained as well as the shortcomings of the study and the future related research outlook in the conclusion.

Response 5: We have revised the introduction and added the scientific innovation and contributions of the paper.

Point 6: The following literature on the application of neural networks is suggested to be cited in the introduction.

Response 6: We cited these two papers in the introduction about describing CNNs and related work.

Reviewer 3 Report

This study aimed to propose a new way of identifying cloud images taken from the ground. We constructed a new dataset of surface shots of clouds, known as the SSC, which was overseen by the Macao Meteorological Society to ensure the quality of the dataset.

I have the following comments should be addressed:

1. The abstract need rewritten which includes the issue of the study, findings and contributions.

2. Keywords are very limited.

3. The introduction need more enhancement through discussing the issues related to cloud and images in cloud layers.

4. The study does not include review of previous studies and then clarify what is the gap in this area.

5. Clarify the methodology in diagram

6. Figure 2a and 2b not clear

Author Response

Point 1: The abstract need rewritten which includes the issue of the study, findings and contributions.

Response 1: Thank you very much for the comment. We have rewritten the abstract which includes the issue of the study, findings and contributions.

Point 2: Keywords are very limited.

Response 2: We have summarized more keywords about this paper.

Point 3: The introduction need more enhancement through discussing the issues related to cloud and images in cloud layers.

Response 3: We included Figure 1 to illustrate that clouds play a significant role in the Earth's water cycle. By doing so, we demonstrate that clouds are a significant element that affects regional climate, thus highlighting the necessity of our study.

Point 4: The study does not include review of previous studies and then clarify what is the gap in this area.

Response 4: In the introductory section, we have reworked the section about related work. It highlights the use of satellite cloud maps as a mainstream method for identifying cloud genera and ongoing research in the field of machine learning for cloud genus identification based on ground images. We conclude by summarizing the current limitations of such research.

Point 5: Clarify the methodology in diagram.

Response 5: We have modified the tables to make them standardized.

Point 6: Figure 2a and 2b not clear.

Response 6: We redraw Figure 2a and 2b to make them clearer.

Round 2

Reviewer 2 Report

The manuscript is suitable for publication and the author has done a great deal to refine it